# Establishing Liposome-Immobilized Dexamethasone-Releasing PDMS Membrane for the Cultivation of Retinal Pigment Epithelial Cells and Suppression of Neovascularization

**DOI:** 10.3390/ijms20020241

**Published:** 2019-01-09

**Authors:** Tzu-Wei Lin, Yueh Chien, Yi-Ying Lin, Mong-Lien Wang, Aliaksandr A. Yarmishyn, Yi-Ping Yang, De-Kuang Hwang, Chi-Hsien Peng, Chih-Chien Hsu, Shih-Jen Chen, Ke-Hung Chien

**Affiliations:** 1Department of Medical Research, Taipei Veterans General Hospital, Taipei 11217, Taiwan; backyard0826@gmail.com (T.-W.L.); g39005005@gmail.com (Y.C.); monglien@gmail.com (M.-L.W.); yarmishyn@gmail.com (A.A.Y.); molly0103@gmail.com (Y.-P.Y.); 2Institute of Pharmacology, School of Medicine, National Yang-Ming University, Taipei 11217, Taiwan; s19609005@gm.ym.edu.tw; 3Facility of Medicine, School of Medicine, National Yang-Ming University, Taipei 11217, Taiwan; sjchen96@gmail.com; 4Department of Ophthalmology, Taipei Veterans General Hospital, Taipei 11217, Taiwan; m95gbk@hotmail.com (D.-K.H.); chihchienym@gmail.com (C.-C.H.); 5Department of Public Health and Institute of Public Health, National Yang-Ming University, Taipei 11217, Taiwan; 6Department of Ophthalmology, Shin Kong Wu Ho-Su Memorial Hospital & Fu-Jen Catholic University, Taipei 11101, Taiwan; chpeng1008@gmail.com; 7Institute of Clinical Medicine, National Yang-Ming University, Taipei 11217, Taiwan; 8Department of Ophthalmology, Tri-Service General Hospital and National Defense Medical Center, Taipei 11490, Taiwan

**Keywords:** age-related macular degeneration (AMD), retinal pigment epithelial (RPE) cells, vascular endothelial growth factor (VEGF), polydimethylsiloxane (PDMS), induced pluripotent stem cells (iPSCs)

## Abstract

Age-related macular degeneration (AMD) is the eye disease with the highest epidemic incidence, and has great impact on the aged population. Wet-type AMD commonly has the feature of neovascularization, which destroys the normal retinal structure and visual function. So far, effective therapy options for rescuing visual function in advanced AMD patients are highly limited, especially in wet-type AMD, in which the retinal pigmented epithelium and Bruch’s membrane structure (RPE-BM) are destroyed by abnormal angiogenesis. Anti-VEGF treatment is an effective remedy for the latter type of AMD; however, it is not a curative therapy. Therefore, reconstruction of the complex structure of RPE-BM and controlled release of angiogenesis inhibitors are strongly required for sustained therapy. The major purpose of this study was to develop a dual function biomimetic material, which could mimic the RPE-BM structure and ensure slow release of angiogenesis inhibitor as a novel therapeutic strategy for wet AMD. We herein utilized plasma-modified polydimethylsiloxane (PDMS) sheet to create a biomimetic scaffold mimicking subretinal BM. This dual-surface biomimetic scaffold was coated with laminin and dexamethasone-loaded liposomes. The top surface of PDMS was covalently grafted with laminin and used for cultivation of the retinal pigment epithelial cells differentiated from human induced pluripotent stem cells (hiPSC-RPE). To reach the objective of inhibiting angiogenesis required for treatment of wet AMD, the bottom surface of modified PDMS membrane was further loaded with dexamethasone-containing liposomes via biotin-streptavidin linkage. We demonstrated that hiPSC-RPE cells could proliferate, express normal RPE-specific genes and maintain their phenotype on laminin-coated PDMS membrane, including phagocytosis ability, and secretion of anti-angiogenesis factor PEDF. By using in vitro HUVEC angiogenesis assay, we showed that application of our membrane could suppress oxidative stress-induced angiogenesis, which was manifested in decreased secretion of VEGF by RPE cells and suppression of vascularization. In conclusion, we propose modified biomimetic material for dual delivery of RPE cells and liposome-enveloped dexamethasone, which can be potentially applied for AMD therapy.

## 1. Introduction

Age-related macular degeneration (AMD) is a leading cause of visual impairment and blindness among people older than 55 years old and is classified into two categories, atrophic (dry) and exudative (wet) AMD [1]. In dry AMD patients, the vision loss is correlated with geographic atrophy of retinal pigment epithelium (RPE) and photoreceptors. So far, no effective drugs or therapeutic strategies exist for treatment of dry AMD, especially for the advanced stage with large area of RPE loss [2]. In wet, or exudative, AMD, the destruction of retinal structure is primarily caused by invasion of abnormal blood vessels from choroid into retina, termed choroid neovascularization (CNV) [3]. Vascular endothelial growth factor (VEGF) is the key regulator of angiogenesis [4], which plays a critical role in the pathogenesis of wet AMD [5]. Thus, the current treatment of wet AMD depends on angiogenesis inhibitors [6], including anti-VEGF agents targeting vessel formation and maturation [7,8,9]. However, the currently used anti-VEGF-based therapy requires monthly intravitreal injections, which increases the risk of infection, retinal detachment and lens damage. Furthermore, this life-long treatment is a big burden due to the medical expenses, and is not a fully curative treatment of wet AMD. Therefore, more recent treatment approaches have included early intervention that precedes RPE cell loss and, in later stages, implantation of stem cell-derived RPE or photoreceptor cells.

Previously, RPE cells differentiated from human embryonic stem cells (hESC-RPE) were implanted into subretinal layer of the patients with dry AMD and Stargardt disease [10]. Even though the adverse effect from implantation was not detected, the restoration of visual function remained unsatisfactory. Afterwards, another report described utilizing autologous induced pluripotent stem cell (iPSC)-derived RPE patch [11]. The implant survived with maintenance; however, visual acuity was not improved and macular edema and clumps of aggregating RPE cells were found.

Along with RPE damage, AMD pathogenesis is also characterized by destruction of Bruch’s membrane (BM), which is an acellular, thin, compact layer of extracellular matrix located between RPE and choroid. BM normally supports the intact and functional cell sheet formation via interaction of RPE cells with extracellular proteins, including laminin, fibronectin, and collagen IV. It also acts as a barrier against CNV invasion by direct binding of pigment epithelium-derived factor (PEDF) to collagen I. Taken together, mimicking of RPE-BM complex may be considered as an approach for wet AMD therapy. In the study in nude rats, it was shown that after subretinal injection, hESC-RPE cells cultured on a synthetic parylene substrate survived longer than hESC-RPE cells in suspension without scaffold [12]. This study emphasized that supportive scaffold is important for cell survival, structure forming and functional improvement of RPE replacement strategy, as it mimics RPE-BM complex structure.

Recently, the engineering of RPE-carrier biomimetic scaffolds and their clinical trials in AMD patients have been reported. For example, an RPE patch was generated by grafting human embryonic stem cell (hESC)-derived RPE monolayer onto a coated synthetic basement membrane [13]. Such RPE patches were implanted into the subretinal space of severe exudate AMD patients and survived for over one year under long-term immunosuppression, leading to improvement of visual acuity [13]. In another study, a novel composite implant of ultrathin synthetic parylene substrate mimicking BM and carrying hESC-RPE was developed [14]. Phase 1 clinical study was conducted in patients with non-neovascular age-related macular degeneration (NNAMD) or dry AMD [14]. No progression of vision loss was found, and the transplanted eyes improved visual acuity [14]. These two reports provide strong evidence on the feasibility and safety of transplantation of RPE-grafted biomimetics as an alternative approach for both dry and wet AMD therapy.

We previously developed non-biodegradable scaffold mimicking BM from FDA-approved polydimethylsiloxane (PDMS) with oxygen plasma modification and laminin coating (PDMS-PmL) [15]. Our results indicated that PDMS-PmL enhanced RPE cell attachment, proliferation, polarization, maturation and specific behavior, such as PEDF secretion, melanosome pigment deposits, and phagocytotic ability. We also demonstrated that RPE cell/photoreceptor precursor multilayers could grow on this scaffold and subretinal implantation in porcine eyes showed long-term biocompatibility [15]. Thus, we suggested that PDMS-PmL could be utilized as an alternative synthetic RPE-BM complex for wet AMD. However, to reduce inflammation, graft rejection and inflammatory angiogenesis after transplantation, anti-inflammatory agents may be considered to be utilized in this strategy. Dexamethasone is an anti-inflammatory drug widely used for inflammation control and recently used in combination with anti-VEGF for wet AMD treatment [15,16]. Furthermore, dexamethasone is able to inhibit angiogenesis, making it a promising therapeutic agent for wet AMD. Therefore, combining the dexamethasone into remodeling of RPE-BM complex can be considered. This necessitates development of suitable delivery system for slow release of dexamethasone. Some previous studies have shown that liposomes enhance drug delivery efficiency for dexamethasone [17,18]. Taken together, in the present study, we purpose the combination of BM mimicking scaffold for cultivation hiPSC-RPE cells and liposomes, carrying dexamethasone as another therapeutic strategy for wet AMD.

## 2. Results

### 2.1. Production and Characterization of Laminin-Modified PDMS Film

In this study, we aimed to generate the artificial BM from FDA-approved PDMS material. As shown in the schematic in Figure 1A, preparation of modified PDMS film consisted of three steps: first, hydrophobic PDMS surface was hydrophilized by treatment with oxygen plasma, followed by silanization with aminopropyltriethoxysilane (APTES), and, finally, crosslinking laminin. In our design of PDMS-based biomimetic, it was imperative that its thickness was comparable with that of natural human BM, which is about 2–4 μm. After we created the PDMS scaffold, we measured its thickness using a surface profiler, and it was shown to be 3.3 μm thick, which satisfied the criterion of the thickness of natural human BM (Figure 1B). Due to the hydrophobic nature of PDMS, cells normally do not attach to its surface readily. Therefore, we subjected PDMS film to oxygen plasma treatment, the procedure that replaces CH3 groups on the surface of the PDMS film with OH, thus increasing hydrophilicity. After that, the water contact angle was measured, and it was shown that after treatment with oxygen plasma, PDMS film had significantly smaller angle of water contact than without treatment, indicating the increased hydrophilicity (Figure 1C). After PDMS surface hydrophilization, we performed its grafting with APTES. To confirm that APTES-treated surface carried the NH2 group, surface analysis by X-ray photoelectron spectroscopy (XPS) was performed. The results indicated that the proportion of nitrogen and carbon elements on the surface increased, on the other hand, the proportion of oxygen decreased after APTES treatment, thus confirming that PDMS surface was successfully modified (Figure 1D). Next, we performed crosslinking of laminin to the NH2 groups introduced by grafting with APTES. For laminin coating, two parameters were optimized: concentration of laminin and proportion of protein crosslinker components (EDC:NHS) used to connect the laminin layer to the PDMS film. The efficiency of laminin coating was quantified by immunofluorescence staining, and it was shown that the amount of laminin crosslinked to the surface treated with 10 μm/mL laminin was significantly higher than to that treated with 5 μm/mL laminin (Figure 1E). At the same time, the ratio EDC:NHS at 4:1 resulted in more efficient laminin crosslinking as compared to a 1:1 proportion (Figure 1E). Collectively, the optimized laminin concentration and EDC:NHS proportion was 10 μm/mL and 4:1, respectively. We also found that coated laminin was stable for at least 120 days (Figure 1F). Finally, in order to investigate the modified PDMS biocompatibility, we cultivated ARPE-19 cell line on its surface and performed a cell viability assay. As shown in Figure 1G, ARPE-19 cells grown on laminin-modified PDMS had significantly higher viability than cells grown on culture dish plastic and on unmodified PDMS.

### 2.2. Human Induced Pluripotent Stem Cell (hiPSC) Culture on Laminin-Modified PDMS

Since our aim was to develop biomimetic scaffold bearing stem cell-derived RPE cell lineage, we first evaluated whether human induced pluripotent stem cells (hiPSCs) could be cultured on laminin-modified PDMS membranes. For this purpose, we first coated PDMS pre-cured polymer onto the surface of a multi-well plate, and then modified it in situ (Figure 2A). hiPSCs were seeded on top of laminin-modified PDMS, and alkaline phosphatase (AP) staining, as well as immunofluorescence staining of specific pluripotency markers, was performed. The hiPSCs demonstrated normal morphology of round-shaped colonies characteristic of stem cells, as well as positive AP staining (Figure 2B). Additionally, stemness-associated markers NANOG, Oct-4 and TRA-1-60 were shown to be normally expressed in hiPSCs cultured on our modified PDMS membrane (Figure 2C).

### 2.3. hiPSC-Derived Retinal Pigment Epithelial (hiPSC-RPE) Cell Growth on Laminin-Modified PDMS

To study the growth of RPE cells on PDMS-coated film, they were differentiated from hiPSCs and grown in a normal cell culture dish and on top of laminin-coated PDMS (Figure 3A). hiPSC-RPE cells were well attached and grew on the modified PDMS scaffold with similar morphology to the cells grown on normal culture dish plastic, including their size, shape, melanin pigmentation and tight junction formation (Figure 3B). Furthermore, hiPSC-RPE cells cultured on modified PDMS film expressed RPE-specific markers, RPE65, BEST1, and ZO1, at comparable levels with the cells grown on plastic, as shown by immunofluorescence staining (Figure 3C). The expression of RPE-specific markers *MITF* and *RLBP1* was confirmed by RT-PCR (Figure 3D), and the markers *RPE65*, *BEST1*, *MITF*, *PAX6* by qRT-PCR (Figure 3E).

### 2.4. RPE Cells Cultured on Laminin-Modified PDMS Film Demonstrate Normal RPE Biological Functions

In normal conditions, RPE cells perform tissue a repairing function when a wound is incurred, which depends on their migration ability. Thus, we performed the wound healing test on hiPSC-RPE cells cultured on modified PDMS scaffold in comparison with hiPSC-RPE cells cultured on tissue culture dish plastic. As shown in Figure 4A, cell migration rate (wound healing ability) of hiPSC-RPE cells on PDMS scaffold was similar to that in a normal cell culture dish. Another important function of normal RPE cells is secretion of the anti-angiogenesis factor PEDF. Therefore, after hiPSC-RPE cells were cultured on laminin-modified PDMS film, we investigated the PEDF secretion by hiPSC-RPE after 3 days, 8 days and 15 days of cultivation by using ELISA. It was shown that the PEDF secretion on day 3 and day 15 was significantly higher in hiPSC-RPE cells cultivated on PDMS than in those grown on tissue culture dish plastic (Figure 4B). Another important characteristic of RPE is phagocytosis capacity. Healthy RPE cells can engulf the metabolites produced by photoreceptors. Herein, we used the pHrodo indicator to determine the phagocytosis ability of hiPSC-RPE cells grown on PDMS compared to the control cells grown on culture dish plastic. As shown in Figure 4C, the phagocytic ability of hiPSC-RPE cells cultivated on laminin-coated PDMS films was comparable to that of cells grown on culture dish plastic. The proportion of P2 population (with phagocytic ability) was 84.4% in the latter and 89.8% in the former case (Figure 4C). To summarize, these results suggested that the hiPSC-RPE cells cultured on laminin-modified PDMS membranes had normal characteristics. 

### 2.5. PDMS Film Biotin Surface Modification and Coating with Dexamethasone-Loaded Liposomes

To generate the dual-function PDMS scaffold that can imitate the subretinal environment and allow the slow release of the anti-angiogenesis drug for wet AMD therapy, we further modified the bottom surface of PDMS film (Figure 5A). The surface was subjected to oxygen plasma treatment, and silane-PEG-biotin was used to conjugate biotin to the surface (Figure 5A). The biotinylated liposomes were loaded with dexamethasome and were attached to the modified PDMS membrane via streptavidin bridges (Figure 5A). To confirm that liposomes were loaded with dexamethasone, we measured the size of liposome particles by using a particle size analyzer. As was previously shown, dexamethasone-loaded liposomes range in size from 100 to 200 nm. Here, we found that the particle size of liposomes without dexamethasone was 100 nm on average; however, after dexamethasone was loaded, the average particle size increased to approximately 150 nm (Figure 5B). These data imply that dexamethasone was successfully encapsulated into liposomes.

### 2.6. The Inhibitory Effect of Dexamethasone Released from Liposomes on Secretion of VEGF by RPE

Oxidative stress is an important factor in the etiology of AMD. Previous reports used tert-butyl hydroperoxide (tBH) to induce the oxidative stress in RPE cells [19]. One of the consequences of tBH-induced oxidative stress in RPE is stimulation of secretion of VEGF, the key factor in angiogenesis [20]. Therefore, in this study, we tested the effect of our membrane on VEGF secretion and associated angiogenesis, thus testing its therapeutic potential for wet AMD treatment. We used an in vitro experimental setup that mimics the processes of VEGF secretion by RPE and neovascularization occurring during AMD in vivo (Figure 6A). In such a setup, the membranes carrying a sheet of hiPSC-RPE cells on the top and either empty or dexamethasone-loaded liposomes on the bottom were placed in the culture dishes containing no tBH (control) or 90 μM tBH (Figure 6A). It was shown that dexamethasone-loaded liposomes demonstrated long-term sustained drug release profile with the first release burst in the initial period of time, and successive slow release behavior (Figure 6B). After 24 h of incubation with tBH, the tBH-containing medium was replaced with fresh medium and further incubated for 24 h to allow accumulation of secreted VEGF (Figure 6A). The ELISA analysis revealed that VEGF secretion level increased by around 40% after tBH stimulation of hiPSC-RPE cells, while dexamethasone treatment significantly lowered VEGF secretion to approximately 30%, as compared to untreated control cells (Figure 6C). Such hiPSC-RPE-conditioned medium was added to human umbilical cord endothelial cells (HUVECs), and in the tube formation assay it was shown that the medium preconditioned by hiPSC-RPE cells treated with tBH caused higher capability of tube formation by HUVECs (Figure 6D). On the other hand, the medium derived from the cells exposed to both tBH and dexamethasone-loaded liposomes caused significantly lower tube formation by HUVECs (Figure 6D). To summarize, our results demonstrate that dexamethasone released from the PDMS-attached liposomes is a potent inhibitor of oxidative stress-induced angiogenesis, which makes such a design a promising approach for therapy of wet-type AMD. 

## 3. Discussion

Age-related macular degeneration (AMD) is a degenerative retinal disease, which is currently the most common cause of visual impairment among the elderly. During the disease progression of dry and wet types of AMD, Bruch’s membrane (BM) undergoes significant age-related changes associated with retinal pigment epithelium (RPE) dysfunction. BM is a thin (2–4 μm) five-layered extracellular matrix, which is composed of the RPE basement membrane, inner collagenous layer, middle elastic layer, outer collagenous layer, and the choroidal endothelial cell basement membrane. BM lies between the RPE and the choriocapillaris, and thus serves two major functions: as the substrate for RPE attachment and as a vessel wall. It supports the intact and functional cell sheet formation via interaction of RPE cells with extracellular proteins, including laminin, fibronectin, and collagen IV. BM also acts as a barrier against choroidal neovascular invasion by direct binding of pigment epithelium-derived factor (PEDF) to collagen type I. BM has been suggested to perform the function of a feeder layer and environment support. Therefore, the artificial BM should fulfill these two requirements of being an RPE substratum and a vessel wall. Numerous studies have investigated the chemical attachment of extracellular matrix (ECM) proteins to the surface of various synthetic polymers in order to provide the long-term mechanical support and ideally mimic the natural microenvironment of RPE cells. Laminins are the major non-collagenous basement membrane components of BM that may play a role in RPE resurfacing. Since BM plays a critical role in maintaining RPE function and preventing neovascularization, reconstruction of BM becomes our major target for the treatment of wet AMD. Herein we demonstrated that a PDMS-based dual-function biomimetic film with laminin-based BM-like scaffold design and RPE cells derived from hiPSCs combined with liposome-enveloped dexamethasone could potentially simulate microenvironment of retina structure to provide a treatment for wet AMD patients. This is the first developed dual function biofilm with a prospect of wet AMD treatment.

PDMS is one of the most extensively characterized synthetic materials for biomedical devices. PDMS is made from stretchable polymer and covers a wide range of physiologically relevant elastic moduli by adjustment of base-to-curing agent ratio. The mechanical properties of PDMS are suitable for manufacturing the custom-designed ultra-thin membranes. PDMS is highly biocompatible and FDA-approved. While natural polymers such as collagen can be advantageous, as they are biodegradable, PDMS provides better opportunity for chemical manipulation aimed at bioengineering, and this feature was used in our study [21]. The main disadvantage of PDMS for mimicking the function of BM is its low permeability for chemical compounds, which can compromise the normal function of BM, which normally allows the nutrient and metabolite flow between the retina and the choroidal blood supply. In our previous study, we have already designed and characterized the BM mimic based on laminin-coated PDMS [15]. Importantly, we have shown in vivo biocompatibility of laminin-modified PDMS by performing subretinal implantation into the pig retina, and have demonstrated that it did not affect the visual function in the course of two years [15]. We hypothesize that the thickness of our membrane, which is about 3 μm and comparable to that of the natural BM, allows the satisfactory flow of chemicals. This hypothesis is corroborated by our previous study, where it was shown that the BM mimic of similar thickness based on parylene, a material with physical properties to comparable PDMS, ensured permeability similar to the natural BM [22]. Additionally, for future application, the architecture of the PDMS membrane can be optimized by introducing the pores as it has been previously demonstrated for polyimide-based membrane [23]. As a continuation of our previous study [15], we utilized the advantageous chemical properties of PDMS to create the dual function membrane, namely, firstly as the substrate for RPE cells, and secondly, as the substrate for vesicles carrying pharmacological agent. For this purpose, we performed chemical modification of PDMS membrane, which included hydroxylation with oxygen plasma, silanization, and, finally, crosslinking the desired compounds: laminin aimed at culturing RPE cells on the one hand, and biotin aimed at attaching drug-loaded liposomes on the other. Thus, we created a dual-function membrane for targeting two major mechanisms of wet AMD: firstly, extensive neovascularization; secondly, replacing dying RPE cells.

Indeed, anti-VEGF treatment is one of the most widely used strategies for wet AMD treatment, and multiple drugs have already been approved [9,20,21,22,23]. On the other hand, replacing or regeneration of lost retinal cells in conditions like AMD with stem cell-based therapies is now a promising therapy method. The autologous RPE sheet implants have passed clinical phase I study in Japan [11]. In our study, we aimed to provide a new approach to combining RPE cell implantation with anti-vascularization treatment. As for the latter, we used liposomes loaded with dexamethasone, which is a corticosteroid compound known for its therapeutic effects in wet AMD [24]. Dexamethasone was shown to reduce proliferation and inhibit migration of retinal endothelial cells [25]. The intravitreal implant with a sustained release of dexamethasone was applied to cull intraocular inflammations [26]. The combination of corticosteroids and anti-VEGF has been reported to perform efficiently in recent AMD studies [27,28].

To summarize, in this study we created the dual function PDMS membrane for potential treatment of wet AMD, designed to deliver drug to suppress neovascularization in combination with mimicking BM for supporting the RPE growth.

## 4. Materials and Methods

### 4.1. PDMS Film Curing

SYLGARD 184 Silicone Elastomer, a two-component silicone kit comprised of base and curing agent, was mixed in a 10(base):1(curing agent) ratio by weight. 10% n-hexane (Seedchem, Melbourne, Australia) was added into the mixture to reduce viscosity. The mixture was then centrifuged at 500 rpm for 1 min after shaking for 1 min and spun in spin-coater for two-stage coating (Stage I: rotation speed: 1200 rpm, acceleration: 600 rpm/s, time: 12 s, Stage II: rotation speed: 3000 rpm, acceleration: 600 rpm/s, time: 63 s).

### 4.2. Film Thickness Measurement

After the film curing process, Microfigure Measuring Instrument (Surfcorder ET-3000, Kosaka Laboratory Ltd., Tokyo, Japan) was utilized to precisely measure the thickness of a film in nanoscale. The prepared samples of different heights were put on the stage. The force of 50 μN was used. The material surface was horizontally scanned at 100 μm/s speed. The thickness of film was calculated from the force change and the height difference.

### 4.3. Surface Hydrophobicity Analysis

The hydrophobicity of oxygen plasma-modified PDMS film was analyzed by contact angle meter (Contact angle model SB). Briefly, the surface samples were set up on a stand. A 5 μL drop of ddH2O was placed on each surface with a titrator. An image of the droplet on the test surface was taken. The analysis software was used to measure the angle of the liquid on the surface.

### 4.4. Surface Modification for Laminin Crosslinking

First, the PDMS film surface was pre-treated by oxidation via oxygen plasma treatment. In brief, the PDMS film surface was exposed to oxygen plasma (Junsun Tech Co., Ltd., New Taipei City, Taiwan) at 10-2 Torr for 5 min at 50 W, and an oxygen flow rate of 17 sccm, in order to create a hydrophilic surface (PDMS-OH). After that, aminization of PDMS surface (PDMS-NH2 was performed by immersing the PDMS film in 1% (*v*/*v*) APTES solution (Sigma-Aldrich, St. Louis, MO, USA) in absolute ethanol, followed by exposure to 5% (*v*/*v*) DI water in ethanol for 15 min at 75 °C in order to hydrolyze the silane. After that, the PDMS sample was washed once with 75% ethanol and three times with DI water to remove residual silane compounds. Finally, PDMS surface was conjugated with laminin by crosslinker 1-ethyl-3-(3-dimethylamino-propyl)-carbodiimide/*N*-hydroxyl-succinimide (EDC/NHS) (Sigma-Aldrich, St. Louis, MO, USA). To obtain a final concentration of 10 mM, EDC/NHS in 1:1 molar ratio was added to 10 μg/mL laminin in PBS buffer and the membrane was incubated for 1 h at 37 °C. Afterward, laminin-grafted PDMS film was washed in DI water to remove residual reagents. Before seeding the cells, PDMS film was rinsed again in PBS. Laminin binding efficiency was quantified by immunofluorescent staining by treating the membrane with anti-laminin primary antibody (ab11575, Abcam, Cambridge, MA, USA) for 4 h at 4 °C, followed by incubation with secondary antibody conjugated with FITC. Images were obtained using fluorescent microscope and a digital camera (DP72, Olympus, Tokyo, Japan) and intensity of fluorescent signal was quantitated using ImageJ software version 1.52a (National Institutes of Health, Bethesda, MD, USA).

### 4.5. Preparation of Dexamethasone-Loaded Liposomes

HSPC (Avanti Polar Lipids, Alabaster, AL, USA), cholesterol (Sigma-Aldrich, St. Louis, MO, USA), DSPE-PEG(2000)-maleimide (Avanti Polar Lipids), DSPE-PEG(2000)-biotin (Avanti Polar Lipids) were mixed at a ratio of 55:40:4:1 and dissolved in chloroform/methanol (10:1). Then, the solution was concentrated in a rotary evaporator at 50 °C and 120 rpm. After concentration, 1 mL of 50 mg/mL dexamethasone was then added and filtered with polycarbonate membrane (Whatman, Maidstone Kent, UK) and stored at 4 °C. For size measurement, 10 µL of liposomes was mixed with 990 µL ddH_2_O, thoroughly mixed and the size was analyzed with NanoBrook 90Plus DLS particle size analyzer (Brookhaven Instruments, Holtsville, NY, USA).

### 4.6. hiPSCs Generation and Cultivation

hiPSCs were generated by nucleofection of peripheral blood mononuclear cells (PBMCs) with reprogramming plasmid mixture through AmaxaTM human T Cell NucleofectorTM Kit (Lonza, Basel, Switzerland). For each nuclofection, cells were transfected with 0.83 μg of PCXLE-hOCT3/4-shp53, PCXLE-hSK, pCXLE-hUL, and 0.5 μg of pCXWB-EBNA1. Transfected cells were then cultivated and refreshed every 10–14 days using freshly thawed inactivated mouse embryonic fibroblasts (MEFs) feeder cells. 3–4 weeks after nucleofection, the number of hiPSC colonies positive for alkaline phosphatase were counted. hiPSCs were maintained on inactivated MEFs (50,000 cells/cm^2^) and fed in human ESC medium (DMEM/F12 (ThermoFisher Scientific, Waltham, MA, USA) supplemented with 20% KnockOut serum replacer (KSR; ThermoFisher Scientific, Waltham, MA, USA), 0.1 mM non-essential amino acids (Invitrogen, Waltham, MA, USA ), 1 mM l-glutamine, 0.1 mM β-mercaptoethanol, 10 ng/mL recombinant human basic fibroblast growth factor (bFGF), and antibiotics (ThermoFisher Scientific, Waltham, MA, USA)). As described previously [29], hiPSCs were then transferred to feeder-free/serum-free culture in HESF V2 medium (Cell Science & Technology Institute, Inc., Sendai, Japan) without KSR supplementation in order to prevent cell contamination by MEFs. 

### 4.7. Differentiation of hiPSCs to RPE Cells

hiPSCs were differentiated from RPE cells as previously described [30]. hiPSCs were passaged onto Geltrex-coated dishes in DMEM/F12 supplemented with N-2 and B-27. From days 0 to 2, this base medium was supplemented with 10 ng/mL IGF1, 50 ng/mL Noggin and 10 ng/mL DKK1. From days 2 to 4 with 10 ng/mL IGF1, 10 ng/mL Noggin, 10 ng/mL DKK1 and 5 ng/mL bFGF. From days 4 to 6 with 10 ng/mL IGF1 and 10 ng/mL DKK1. From days 6 to 8 with 100 ng/mL Activin A (R&D systems, Minneapolis, MN, USA) and 10 μM SU5402. From days 6 to 14 with 100 ng/mL Activin A. On day 14, the foci of the melanin-stained differentiated RPE cells were manually separated from undifferentiated cells and passaged onto a new dish. One more passage was repeated, and on day 40, the RPE cells were harvested for RNA extraction and microarray analysis.

### 4.8. tBH-Treated RPE-Conditioned Medium HUVEC Assay

The PDMS blocks with a size of 10 mm × 10 mm × 2 mm (L × W × H) were placed at the bottom of a 6 cm culture dish with a distance between them of 5 mm. The PDMS membranes were placed across the top of these blocks and fixated with staples on each side. The fixed PDMS membrane on which RPE cells were cultured was immersed into the medium, and 90 uM tBH was added accordingly. The RPE-conditioned medium was added to HUVEC culture on 96-well plate and examined after 18 h for tube and segment lengths.

### 4.9. Alkaline Phosphatase Assay

The hiPSCs were fixed with 4% paraformaldehyde and washed with PBST. The alkaline phosphatase activity was determined by the Alkaline Phosphatase Substrate Kit III (Vector Labs, Burlingame, CA, USA) following the manufacturer’s protocol. Colonies stained red indicated positive alkaline phosphatase activity.

### 4.10. Reverse Transcription-Polymerase Chain Reaction (RT-PCR)

Total RNA was isolated by using TRIzol Reagent (Invitrogen, Waltham, MA, USA) prior to quantification by spectrophotometry. To synthesize first-strand cDNA, 5 μg of each total RNA was reverse-transcribed with SuperScript III (Invitrogen, Waltham, MA, USA) on GeneAmp PCR System 9700 thermocycler (Applied Biosystems, Waltham, MA, USA). The PCR reactions were performed as follows: pre-denaturation at 94 °C for 5 min, followed by 25 to 30 cycles of denaturation at 94 °C for 30 s, annealing at 58–62 °C for 30 s, and extension at 72 °C for 45 s. PCR products were loaded on 1% agarose gels and visualized using ethidium bromide staining and a camera system (Transilluminator/SPOT; Diagnostic Instruments, Inc., Sterling Heights, MI, USA). The gel images of the RT-PCR products were directly scanned (ONEDscan 1-D Gel Analysis Software; Scanalytic Inc., Milwaukee, WI, USA).

### 4.11. Quantitative RT-PCR (qRT-PCR)

The amplification was carried out in a total volume of 20 μL, containing 0.5 mM of each primer, 4 mM MgCl_2_, 20 μL LightCycler FastStart DNA Master SYBR Green I (Roche Diagnostics, Basel, Switzerland) and 20 μL of 1:10 diluted cDNA. The quantification of the unknown samples was performed by LightCycler Relative Quantification Software, version 3.3 (Roche Diagnostics, Basel, Switzerland). PCR reactions were prepared in duplicate and heated to 95 °C for 10 min followed by 40 cycles of denaturation at 95 °C for 10 s, annealing at 55 °C for 15 s, and extension at 72 °C for 20 s. Standard curves (cycle threshold values versus template concentration) were prepared for each target gene and for the endogenous reference (GAPDH) in each sample.

### 4.12. Immunofluorescence Staining

The target cells were collected and then fixed in 4% paraformaldehyde. Cell permeabilization was conducted in 0.1% Triton X-100 prior to blocking in 5% normal goat serum in PBS. Cell samples were then incubated with primary antibodies. After three washes in PBS, the cells were then incubated with goat anti-mouse or secondary antibodies conjugated with FITC (green) or PE (red). Cell nuclei were stain with DAPI (blue). Images were obtained using fluorescent microscope and a digital camera.

### 4.13. MTT Assay

2 × 10^4^ cells were seeded into each well of a 24-well plate. Methyl thiazol tetrazolium (MTT; Sigma-Aldrich, St. Louis, MO, USA) was added appropriately, and cells were incubated at 37 °C for two more hours. The medium was removed and DMSO was added. The MTT formazan product was quantified using a microplate reader at absorbance of 560 nm (SpectraMax 250, Molecular Devices, San Jose, CA, USA).

### 4.14. Enzyme-Linked Immunosorbent Assay (ELISA)

After 3 days of RPE cells incubation on different surfaces, culture medium was collected and measured for PDGF-BB (ab100624; Abcam, Cambridge, MA, USA) according to the manufacturer’s protocol. The optical densities were determined within 30 min and recorded with a microplate reader (EXL800; BioTek, Winooski, VT, USA) at 450 nm.

### 4.15. Phagocytosis Assay

hiPSC-RPE cells were seeded at different conditions and cultured overnight. pHrodo Green Intracellular pH Indicator (Thermo Fisher Scientific, Waltham, MA, USA) was added to the RPE cells to observe the phagocytosis-induced intracellular pH change, according to the manufacturer’s instruction. After incubation at the indicated concentrations and time, cells were collected and fixed. FITC signal intensity was measured during flow cytometry analysis.

### 4.16. Wound Healing Assay

hiPSC-RPE cells were seeded on different surfaces and allowed to grow to confluence.

A scratch wound was produced on each cell surface by using a micropipette tip. The wound area was photographed at different time points: 0, 12, 30, and 36 h. The lines of the wound edges were made manually, and the distance between the two edge lines was measured.

### 4.17. Statistical Analysis

The results are shown as mean ± SD. Statistical analyses were performed using the *t*-test for comparison between 2 groups, and one-way ANOVA for among 3 or more groups. The results were considered statistically significant at *p* < 0.05. 

## 5. Conclusions

To summarize, this is the first study where we created the dual function PDMS membrane for potential treatment of wet AMD, aimed at both suppressing neovascularization and replacement of the lost RPE cells.

## Figures and Tables

**Figure 1 ijms-20-00241-f001:**
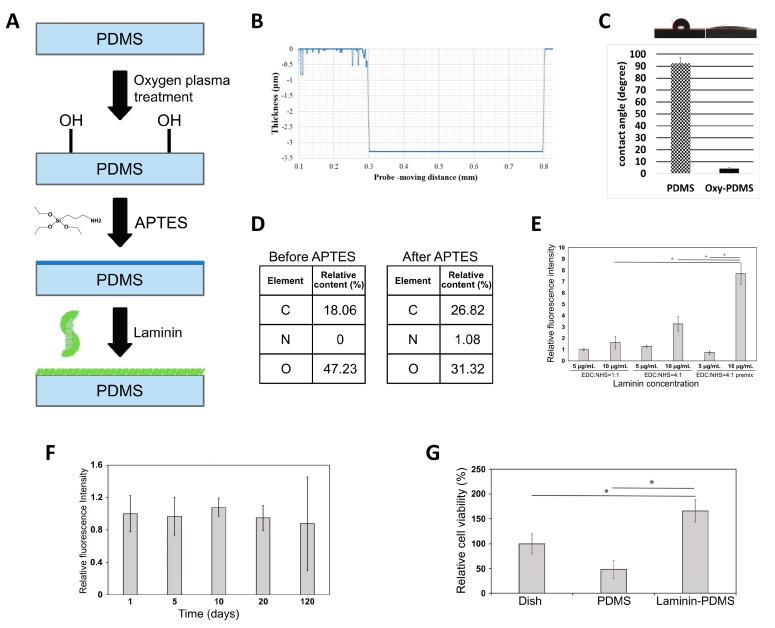
Production and characterization of laminin-modified PDMS film. (**A**) Schematic representation of the manufacturing process of laminin-coated PDMS, consisting of stages of hydrophilization with oxygen plasma, silanization with APTES, and crosslinking laminin. (**B**) Measurement of the thickness of the PDMS membrane by surface profiler. (**C**) Water contact angle measurement before (left) and after (right) oxygen plasma treatment. (**D**) X-ray photoelectron spectroscopy analysis, showing increased carbon and nitrogen elements in PDMS due to APTES grafting. (**E**) Optimization of laminin concentration and crosslinking agent ratio (EDC:NHS) used for laminin crosslinking. (**F**) Stability of laminin crosslinked to PDMS membrane in a course of 120 days measured by immunofluorescence assay. (**G**) Cell viability assay performed on ARPE-19 cells grown on laminin-modified PDMS as compared to uncoated laminin and culture dish plastic. The values in (**E**–**G**) are the means from three independent measurements with SD error bars, * indicates statistically significant difference (ANOVA, *p* < 0.05).

**Figure 2 ijms-20-00241-f002:**
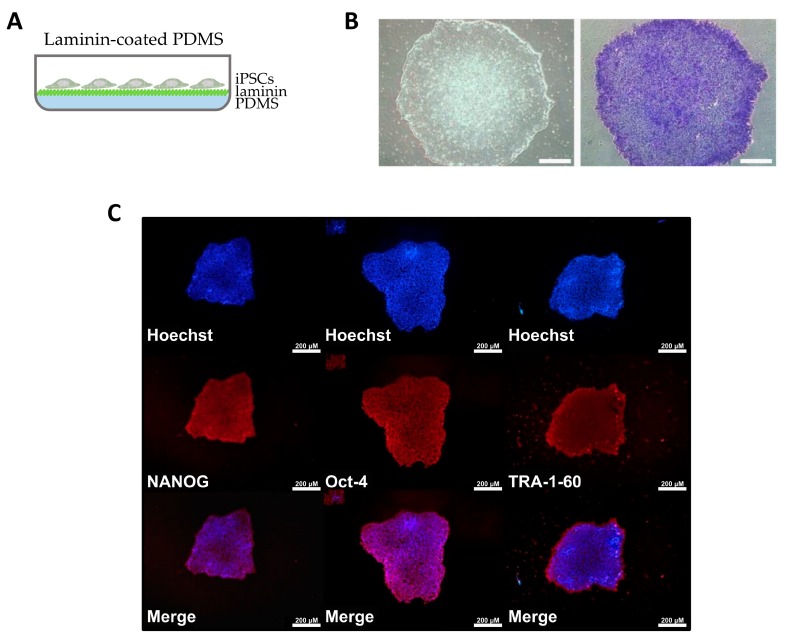
hiPSC culture on laminin-modified PDMS. (**A**) Schematic showing cultivation of hiPSCs in a dish with laminin-coated PDMS. (**B**) Bright-field images of hiPSCs grown on top of laminin-coated PDMS without (left) and with (right) alkaline phosphatase staining. Scale bar = 200 µm (**C**) Immunofluorescence staining of pluripotency markers in hiPSCs grown on laminin-coated PDMS. Nuclei stained with Hoechst dye (Thermo Fisher Scientific, Waltham, MA, USA).

**Figure 3 ijms-20-00241-f003:**
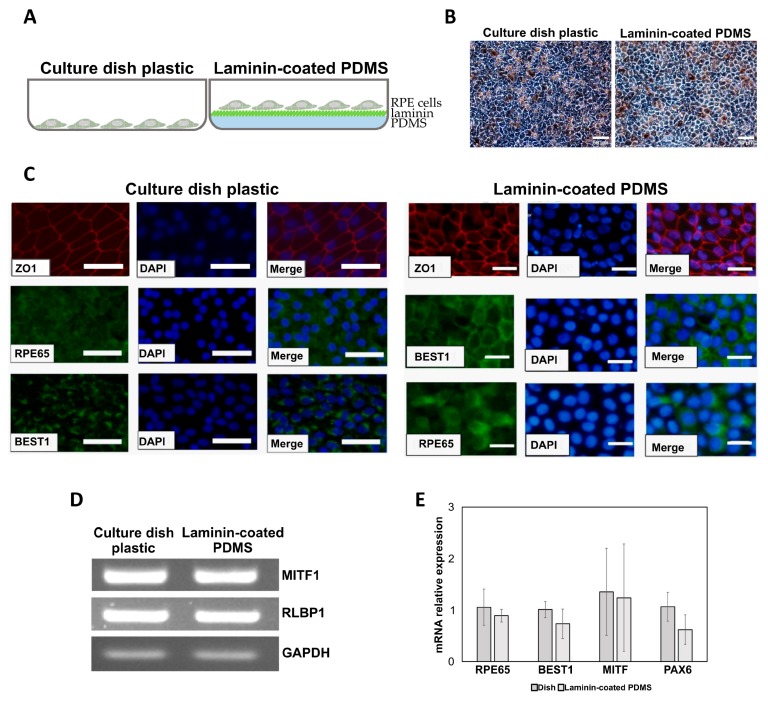
hiPSC-RPE cell growth on laminin-modified PDMS. (**A**) Schematic showing cultivation of hiPSC-RPE cells on tissue culture dish plastic control (left) and in a dish with laminin-coated PDMS (right). (**B**) Bright-field images of hiPSC-RPE cells grown in a dish (left) or on laminin-PDMS (right). Scale bar = 50 µm (**C**) Immunofluorescence staining of typical RPE markers. DAPI- nuclear stain. Scale bar = 100 µm (**D**) RT-PCR analysis of expression of RPE markers *RLBP1* and *MITF1*. *GAPDH* mRNA was detected as a loading control. (**E**) qRT-PCR analysis of expression of the indicated RPE markers. Expression levels in hiPSC-RPE cells quantified relative to expression levels in cells grown on plastic culture dish. The relative levels are the means from three independent samples with SD error bars.

**Figure 4 ijms-20-00241-f004:**
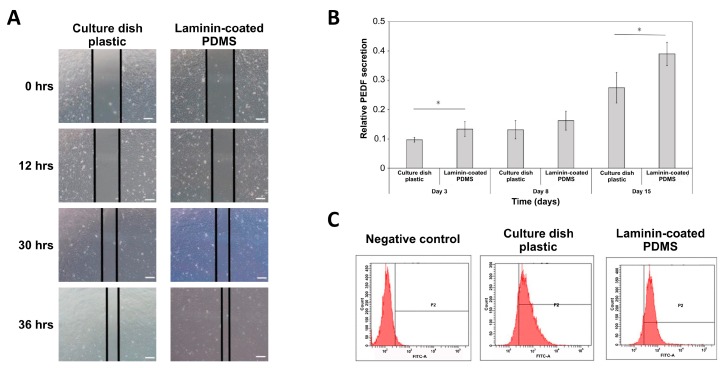
RPE cells cultured on laminin-modified PDMS film demonstrate normal RPE biological functions. (**A**) Cell migration (wound healing) assay performed with hiPSC-RPE cells grown on plastic and laminin-modified PDMS. Scale bar = 100 µm (**B**) PEDF secretion by hiPSC-RPE cells grown on plastic and laminin-PDMS analyzed by ELISA assay in a time course of 3, 8 and 15 days. The values are the means from three independent measurements with SD error bars, * indicates statistically significant difference (ANOVA, *p* < 0.05). (**C**) Flow cytometry analysis of phagocytosis in populations of hiPSCs-RPE cells grown on plastic and laminin-PDMS.

**Figure 5 ijms-20-00241-f005:**
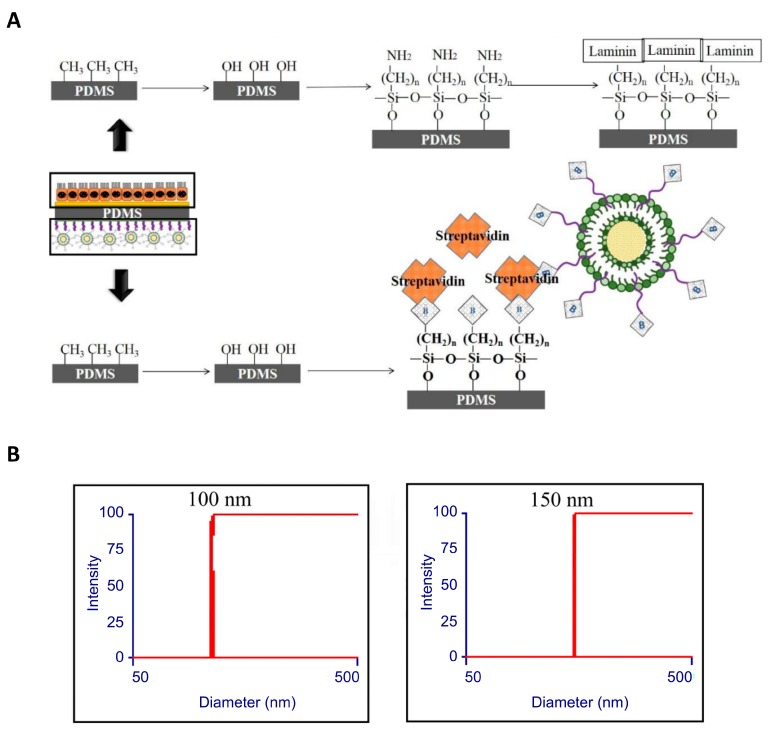
Preparation of modified PDMS membrane with attached dexamethasone-loaded liposomes. (**A**) Schematic diagram showing the process of preparation of liposome-loaded PDMS membrane. The top surface coated with laminin via the steps of hydroxylation, silanization and laminin crosslinking; the bottom surface loaded with liposomes via hydroxylation, silane-biotinylation, and noncovalent attachment of biotinylated liposomes via streptavidin bridges. (**B**) Liposome diameter measurement before (left) and after (right) loading with dexamethasone.

**Figure 6 ijms-20-00241-f006:**
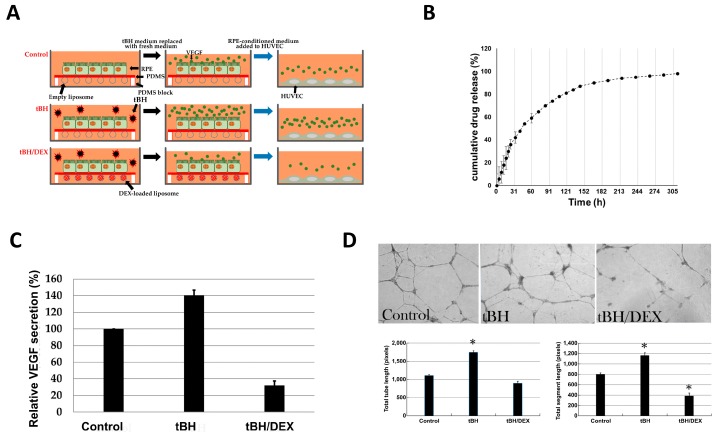
The inhibitory effect of dexamethasone (DEX) released from liposomes on tBH-induced angiogenesis. (**A**) Schematic representation of the experimental setup. (**B**) Dexamethasone-loaded liposome drug release profile, showing sustained drug release. (**C**) ELISA assay showing secretion of VEGF by hiPSC-RPE after stimulation with tBH. The values are the means from three independent measurements expressed as a percentage relative to the control with SD error bars. (**D**) The tube formation assay on HUVECs after stimulation with RPE-conditioned media as shown in (**A**). Representative bight-field images shown at the top, quantification of total tube length (left) and total segment length (right)—at the bottom. Original magnification, × 20. In (**C**,**D**), control—untreated HUVECs, tBH—HUVECs treated with tBH, tBH/DEX—HUVECs pretreated with tBH followed by exposure to an insert with the dexamethasone-loaded liposomes. The values in (**B**–**D**) are the means from three independent measurements with SD error bars. * *p* < 0.05 Student’s *t*-test, vs. control.

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
