# Peer review of "Establishing Liposome-Immobilized Dexamethasone-Releasing PDMS Membrane for the Cultivation of Retinal Pigment Epithelial Cells and Suppression of Neovascularization"

_ijms, 2019, doi:10.3390/ijms20020241_

Round 1
Reviewer 1 Report
The manuscript has substantially improved by the revisions. Still, there are some aspects to be considered before publication.
Abstract
In their abstract the authors claim that there is no effective therapy for wet AMD. This is simply not correct as anti-VEGF treatments is effective, as shown by several clinical trials. It is not a curative therapy, but effective.
Figures
Please include the numbers of experiments (n=)
Discussion
The discussion has been a bit too much curtailed. Please discuss the results (and make clear what is additional to the Oncogene study) in an appropriate manner
Line 319, I do not understand how a laminin-covered membrane targets the death of RPE cells.
Style
Page 12, line 371-373, the font has changed
Author Response
1. Abstract
In their abstract the authors claim that there is no effective therapy for wet AMD. This is simply not correct as anti-VEGF treatments is effective, as shown by several clinical trials. It is not a curative therapy, but effective.
Reply: Thanks for noticing this inaccuracy. We have modified this sentence accordingly: “So far, the effective therapy options for rescuing visual function in advanced AMD patients are largely limited, especially in wet type AMD, in which retinal pigmented epithelium and Bruch’s membrane structure (RPE-BM) is destroyed by abnormal angiogenesis. Anti-VEGF treatment is effective remedy for the latter type of AMD, however, is not a curative therapy. Therefore, …”
2. Figures
Please include the numbers of experiments (n=).
Reply: The number of experiments was included in the figure legends, wherever it was absent.
3. Discussion
The discussion has been a bit too much curtailed. Please discuss the results (and make clear what is additional to the Oncogene study) in an appropriate manner.
Reply: We have added the Discussion section describing our previous study in Oncotarget and stating that this paper is its continuation (lines 317-330). We emphasize that in this paper, as compared to Oncotarget paper, we design dual function membrane. On the other hand, in our previous study, we tested biocompatibility of such membrane on pigs.
4. Line 319, I do not understand how a laminin-covered membrane targets the death of RPE cells.
Reply: We have changed it to “… secondly, replacing dying RPE cells.”
5. Style
Page 12, line 371-373, the font has changed.
Reply: Thanks for noticing it. The font has been corrected.

Reviewer 2 Report
The manuscript by Lin et al develops a new scaffold for RPE culture. Authors use a PDMS based scaffold that is functionalized using plasma treatment and laminin coated before RPE culture. Authors show that both iPSCs and RPE cells grown well on this scaffold. One of the main concerns with this manuscript is the utility of PDMS as a scaffold for transplantation of RPE monolayer. PDMS is hydraulically non-conductive. Therefore, it will impede in nutrient and metabolite flow from and to the choroidal blood supply, starving the RPE and the photoreceptors. Authors should test the transplant-ability of their scaffold. Please address these additional comments.
1) Figure 3, quality of RPE cultures is not visible in these images. Please provide higher mag images of RPE cells where RPE hexagonal morphology and pigmentation is clear.
2) All of the analysis authors present in this manuscript is based on relative difference between RPE grown on cell culture plastic and PDMS. The quality of RPE derived from these iPSCs is not clear. Can authors compare their data to primary RPE?
3) Phagocytosis assay should be performed using photoreceptor outer segments (commercially available)
4) There is no statistical analysis of data in Figure 6D
Author Response
Comments and Suggestions for Authors
The manuscript by Lin et al develops a new scaffold for RPE culture. Authors use a PDMS based scaffold that is functionalized using plasma treatment and laminin coated before RPE culture. Authors show that both iPSCs and RPE cells grown well on this scaffold. One of the main concerns with this manuscript is the utility of PDMS as a scaffold for transplantation of RPE monolayer. PDMS is hydraulically non-conductive. Therefore, it will impede in nutrient and metabolite flow from and to the choroidal blood supply, starving the RPE and the photoreceptors. Authors should test the transplant-ability of their scaffold. Please address these additional comments.
Reply: Thank you, indeed this is the valid concern that PDMS may impede the nutrient flow between the retina and subretinal tissues. However, in our previous study we have demonstrated that implantation of such device into the pig retina did not affect the visual function in the time course of two years. We may hypothesize that PDMS membrane of which is very thin may allow some transport of the molecules, as it has been previously demonstrated in the case of parylene, the material of similar permeability to PDMS [Lu B, Tai YC, Humayun MS. Microdevice-based cell therapy for age-related macular degeneration. Dev Ophthalmol. 2014;53:155-66.]. Additionally, the architecture of such membrane can be optimized to contain pores. We have added these considerations to the Discussion.
1. Figure 3, quality of RPE cultures is not visible in these images. Please provide higher mag images of RPE cells where RPE hexagonal morphology and pigmentation is clear.
Reply: We updated Figure 3 by adding higher magnification image of RPE cells.
2. All of the analysis authors present in this manuscript is based on relative difference between RPE grown on cell culture plastic and PDMS. The quality of RPE derived from these iPSCs is not clear. Can authors compare their data to primary RPE?
Reply: We agree that ideally the data should be compared to primary RPE. However, in this study we demonstrate that iPSC-derived RPE cells have typical morphology, express characteristic markers and demonstrate functionality of normal RPE cells (phagocytosis, PEDF secretion). This data definitely validate the identity of RPE cells.
3. Phagocytosis assay should be performed using photoreceptor outer segments (commercially available)
Reply: Yes, we agree that such experiment can be considered. However, this experiment is not the main focus of this paper, as it was a part of validation to confirm cell identity of iPSC-derived cells.
4. There is no statistical analysis of data in Figure 6D
Reply: The Figure 6D has been updated with statistical data added.